# Addressing a critical need: A randomised controlled feasibility trial of acceptance and commitment therapy for bariatric surgery patients at 15–18 months post-surgery

Elizabeth A. Barley[1]*, Marie Bovell[1], Kate Bennett-Eastley[2], John Tayu Lee[3], Dayna Lee-Baggley[4], Simon S. Skene[2], Michael Z. Tai[5], Sue Brooks[1], Samantha Scholtz[6]

1 School of Health Sciences, University of Surrey, Guildford, United Kingdom, 2 School of Biosciences and Medicine, University of Surrey, Guildford, United Kingdom, 3 School of Population and Global Health, University of Melbourne, Parkville, VIC, Australia, 4 Faculty of Medicine, Dalhousie University, Halifax, Canada, 5 Department of Psychiatry, St Charles Hospital, London, United Kingdom, 6 Division of Diabetes, Endocrinology and Metabolism, Imperial College, London, United Kingdom

* e.barley@surrey.ac.uk

**Data Availability Statement:** All relevant data are within the manuscript and its Supporting Information files.

## Abstract

Bariatric surgery is an effective treatment for obesity. However, around one in five people experience significant weight regain. Acceptance and Commitment Therapy (ACT) teaches acceptance of and defusion from thoughts and feelings which influence behaviour, and commitment to act in line with personal values. To test the feasibility and acceptability of ACT following bariatric surgery a randomised controlled trial of 10 sessions of group ACT or Usual Care Support Group control (SGC) was delivered 15–18 months post bariatric surgery (ISRCTN registry ID: ISRCTN52074801). Participants were compared at baseline, 3, 6 and 12 months using validated questionnaires to assess weight, wellbeing, and healthcare use. A nested, semi-structured interview study was conducted to understand acceptability of the trial and group processes. 80 participants were consented and randomised. Attendance was low for both groups. Only 9 (29%) ACT participants completed > = half of the sessions, this was the case for 13 (35%) SGC participants. Forty-six (57.5%) did not attend the first session. At 12 months, outcome data were available from 19 of the 38 receiving SGC, and from 13 of the 42 receiving ACT. Full datasets were collected for those who remained in the trial. Nine participants from each arm were interviewed. The main barriers to group attendance were travel difficulties and scheduling. Poor initial attendance led to reduced motivation to return. Participants reported a motivation to help others as a reason to join the trial; lack of attendance by peers removed this opportunity and led to further drop out. Participants who attended the ACT groups reported a range of benefits including behaviour change. We conclude that the trial processes were feasible, but that the ACT intervention was not acceptable as delivered. Our data suggest changes to recruitment and intervention delivery that would address this.

**Funding:** This project is funded by the National Institute for Health Research (NIHR) under its Research for Patient Benefit (RfPB) Programme (Grant Reference Number PB-PG-0816-20012). The views expressed are those of the authors and not necessarily those of the NIHR or the Department of Health and Social Care. The grant was awarded to EB as the chief investigator and JTL, SS, GH, and SS as co-investigators. website found: https://fundingawards.nihr.ac.uk/award/PB-PG-0816-20012 The funders had no role in study design, data collection and analysis, decision to publish, or preparation of the manuscript.

**Competing interests:** The authors have declared that no competing interests exist.

## Introduction

Across the world there are over a billion individuals living with obesity (13% prevalence of a BMI >30kg/m2) [1]. This is associated with a host of negative health outcomes, including a 48% increased risk of mortality following COVID-19 infection for individuals living with obesity compared to individuals without obesity (pre-vaccination rates) [2]. Within the UK, 28% of the population is living with obesity and obesity was a primary or secondary factor in over one million admissions to NHS hospitals in the year 2019/2020 [3]. Bariatric surgery is highly effective in treating obesity and its associated comorbidities; in patients with a BMI of over 35 it is more effective than nonsurgical interventions in bringing about weight loss [4]. However, a 2021 systematic review found that 17.6% of patients still experienced significant weight regain of 10% or more [5], which may necessitate further invasive procedures (bariatric surgery costs upwards of £10,000) and a return of comorbid conditions [6, 7].

The UK's National Institute for Health and Care Excellence recommends that bariatric surgery should be supplemented by psychological support, though it does not specify a particular therapy modality [8]. Cognitive behavioural interventions have been found to be effective for weight loss [9] However, a recent systematic review [10], which found only nine randomised controlled trials of the effectiveness of psychological intervention provided to bariatric surgery patients, concluded that there is a lack of evidence in this population. High-quality studies are therefore needed, not least to understand the mechanisms through which cognitive and behavioural therapies may aid bariatric surgery patients.

Analysis of the psychological profile of patients living with obesity found poor self-regulation and negative mood as the strongest predictors of variance in weight loss [11]. Eating psychopathology was associated with a 2.2 odds ratio in the weight regain group compared to the non-weight regain group in a recent systematic analysis [12]. A longitudinal study of over 400 bariatric surgery patients found that, among others, depression, increased hedonic salience of food cues, and distress intolerance predicted poorer weight loss [13]. These mechanisms can be targeted by third wave cognitive behavioural therapies (3wCBT) which use principles of mindfulness and non-judgemental awareness.

Support for this comes from a recent systematic review [14] (n = 37 studies) which found 3wCBT to be more effective for weight loss than standard behavioural treatment; the most consistent evidence was for Acceptance and Commitment Therapy (ACT). That review excluded studies of bariatric patientsbut another review focused on this population ([15] n = 10 studies) found evidence for acceptance-based approaches in supporting wellbeing. Only one randomised controlled trial of ACT in bariatric patients [16] (n = 39) was identified, in which those receiving six weeks of ACT demonstrated greater improvements in disordered eating, body dissatisfaction, QoL and acceptance of weight related thoughts and feelings than those receiving treatment as usual, with effects consistent at 6 months post intervention [16, 17]. However, the participants varied in their post-surgery time, ranging from 4–38 months, with a mean of 15.5 so the ideal timing of psychological intervention is unknown [18]. As the mechanisms hypothesized to drive weight gain will take time to express as new obesogenic behaviours to target in therapy [19], therapies targeting post bariatric surgery patients may be more beneficial if given at a later stage. Multiple studies have established that 20–30% of patients regain weight within 24 months of surgery, unfortunately this is also when patients may be discharged from secondary care [20]. The present feasibility RCT was designed to determine whether ACT delivered in this window, 15–18 months post-bariatric surgery, is acceptable to patients, and to guide the methodology of a future trial assessing effectiveness.

In previous studies of ACT for weight management [17, 21], a range of outcomes, such as disordered eating, cravings, body dissatisfaction, quality of life and weight loss have been used

with alterations in psychological flexibility, a core component of ACT, was suggested to be the primary mediator of benefits in the one RCT. However other mediators for weight loss have been proposed [18], so a wide range of treatment outcomes will be measured and tested for potential mediators and moderators of effect.

Despite guidelines recommending psychological interventions in post-operative care, routine clinical practice has not caught up [22], and evidence to guide clinicians in the type of therapy is limited. If evidence based interventions were available, this support could reduce the costs to the patient and society associated with weight regain and its sequelae. The aim of this feasibility RCT is to determine the acceptability of ACT group therapy in post-bariatric surgery patients and to provide insight into the optimal methods for an RCT assessing the effectiveness of ACT in this patient population. Acceptability of the intervention was assessed as group attendance and through interview data.

## Methods

The full protocol for this trial has been published in the International Journal of Clinical Trials [23]. The supporting CONSORT checklist [24] is available as supporting information; see S1 Fig.

### Design

This was a feasibility study for a single-centre, parallel-group, single-blind, two-arm trial, with randomisation to either a 10 weeks ACT group therapy or Usual Care Support Group control (SGC). Favourable ethical opinion was obtained from London-Westminster REC (18/LO/ 1256). The trial was registered at Researchregistry.com, UIN: 3959 (date registered: 10 April 2018); ISRCTN registry ID: ISRCTN52074801.

### Participants

All participants were adults (≥18years) post bariatric-surgery (>15 to < 18 months) at a London-based weight management centre. Patients who expressed suicidal ideation (score >0 on Q9 of thePHQ-9 [25] or at psychiatry assessment), or who were unable to communicate in English, or who were unable to commit to attendance were excluded.

Initially, potentially eligible patients were given a Participant Information Sheet (PIS) and PHQ-9 at their regular post-surgical follow up appointments, but due to reduced operating capacity in the previous year, recruitment was hampered by low numbers in follow up clinics. Therefore, all eligible patients were contacted and posted or emailed the PIS. Interested patients were approached by the RA and an appointment made to obtain consent and baseline data. Written consent was obtained for all participants. Recruitment ceased when the target was reached.

### Randomisation

This was performed by an independent researcher running R program scripts [26]. The randomisation method uses permuted blocks of variable size, randomly varying from two to six participants. All eligibility criteria were checked prior to randomisation. Each patient was given a unique participant number which was sent to the researcher running the R program. The program generated a randomised patient study number assigning them to either the ACT or SGC. This was emailed back to SS who then informed the participant of their allocation and provided details of where and when to attend. Other team members, including the statistician, were blind to allocation, but the patients knew to which arm they had been randomised. Approximately equal numbers were allocated to each condition.

## Sample size

We aimed to recruit 58 participants. However, due to high attrition, including before the first session, we ran extra groups (i.e. 4 iterations rather than 2) and recruited 80 participants. Loss to follow up and group attrition are presented in a consort diagram (Fig 1).

As this was a feasibility study, we did not carry out a formal sample size calculation when planning the study. However, we provide an estimate of the precision of the primary endpoints (Table 1)–(1a) proportion of eligible patients at screening (1b) proportion of patients agreeing to take part in the study (2) proportion of patients remaining in the study at 12 months (3) completeness of data collection at 12 months and (4) attendance at therapy sessions, based on the recruitment of 80 patients. As an indication of power, recruitment of a target of 80 participants would allow estimation of the retention rate at any period within +/- 11.5%.

## Intervention

This comprised ACT group therapy delivered weekly (10 x 90-minute sessions) by a psychologist trained in bariatric psychological support and ACT. The intervention was adapted from an existing manual which addressed all six core processes of ACT: acceptance, defusion, contacting the present moment, self as context, committed action, values awareness [27]. ACT supervision was provided for the psychologist and a sample of sessions observed for fidelity.

## Control (usual care)

This comprised the support group control (SCG) (10 x 90-minute meetings), delivered weekly by a psychology graduate with no formal therapy training. The groups were unstructured to allow patients to talk freely. The groups were held at the same time and venue, but on a different day of the week to the ACT groups.

## Outcomes

Primary outcomes related to the feasibility of the trial. Measures of the success of recruitment, data collection and acceptability of trial procedures included (1) willingness of participants to be randomised, i.e. the number of participants who drop out of the study after they had been randomised, (2) number of eligible patients attending clinic i.e. the percentage of patients who met eligibility criteria at screening, (3) follow-up rates, i.e. the percentage of participants retained until 12 month follow-up and providing complete datasets, (4) adherence to the intervention or control i.e. number of sessions attended, (5) reasons for attrition, (6) time needed to collect data.

Secondary outcomes explored related to obesity, health and wellbeing. The following validated measures were used: King's Obesity Staging Criteria [28], ICEpop CAPability measure for Adults (ICECAP-A) [29], International Physical Activity Questionnaire (IPAQ) [30], Brief Mediterranean Diet Questionnaire [31], Alcohol Use Disorders Identification Test (AUDIT) [32], Dutch Eating Behaviour Questionnaire (DEBQ) [33], Hospital Anxiety and Depression Scale (HADS) [34], Weight in kilograms and ACT processes using: Distress Tolerance Scale (DTS) [35], Philadelphia Mindfulness Scale (PHLMS) [36], Drexel Defusion Scale (DDS) [37], Physical Activity Acceptance Questionnaire (PAAQ) [38], and Food Craving Acceptance and Awareness Questionnaire (FAAQ) [39].

Economic costs were calculated from the perspective of the NHS. Annual staff training budgets typically fund ACT training, which costs £450 (ranging from £350 to £550) for a healthcare professional with an existing psychology qualification. The ACT group was led by a Band 7 psychologist as part of their regular duties. A Band 4 psychology graduate with no formal

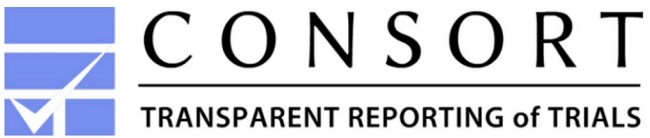

## CONSORT 2010 Flow Diagram

**Enrollment**

Assessed for eligibility
(n=264)

Excluded (n=184)
- Not meeting inclusion criteria (n=18)
- Declined to participate (n=129)
- Baseline interviews booked DNA (n=37)

Randomised (n= 80)

**Allocation**

Allocated to ACT (n=42)
- Received allocated intervention (n=42)

Allocated to SCG (n=38)
- Received allocated intervention (n=38)

**Follow-Up**

Still in study at 3 months n=34
Still in study at 6 months n=32
Still in study at 12 months n=31

Follow-up data collected

3 months n=12

6 months n=14

12 months n=13

Still in study at 3 months n=37
Still in study at 6 months n=37
Still in study at 12 months n=37

Follow-up data collected

3 months n=15

6 months n=23

12 months n=19

**Fig 1. CONSORT flow diagram.** ACT, acceptance and commitment therapy; SCG, support group control.

**Table 1. Data to inform sample size of a future trial.**

| Endpoint | n/N | Proportion (95% confidence interval) |
|---|---|---|
| Eligible at screening | 246/264 | 93% (80% to 96%) |
| Consented to take part | 80/246 | 33% (27% to 39%) |
| Remained in the study @ 12months | 68/80 | 85% (75% to 92%) |
| Completeness of data collection @ 12months* | 32/68 | 47% (35% to 60%) |
| Attendance at therapy sessions** | 18/68 | 26% (17% to 39%) |

*The denominator here is the number of patients still in the study at 12 months

**The numbers attending the therapy sessions were averaged over all cohorts, both randomisation groups and all 10 sessions, although it has already been noted that the number attending each session decreased as the study progressed.

therapy training led the support group. The CSRI was used to track healthcare resource use during the trial period [40]. Using CSRI data, service utilisation costs for 2019/2020 were calculated by multiplying resource use by the appropriate unit cost. For service delivery, such as hospital stay costs, we also used national standard tariffs. Unit costs from the Unit Cost of Health and Social Care 2020 and NHS Reference Costs 2020 were used in the analysis. The EQ-5D was used to calculate quality-adjusted life years (QALY [41]).

Outcome measures were completed at baseline, 3, 6 and 12 months follow up, except for weight and the King's Obesity Staging Criteria [29] which were collected at baseline and 12 months only. Demographic data were collected at baseline. Baseline data were collected face to face at routine clinic appointments; 3- and 6-month FU data were collected via telephone. We planned to collect 12-month data face to face, but due to the global pandemic this was mostly collected by phone.

Participants from both arms were invited to be interviewed after the 12 month follow up in order to understand acceptability of the trial and group processes. Recruitment stopped once equal numbers from each arm had been recruited and the team agreed that data saturation had been achieved. The ACT group participants also completed further ACT process measures at each session as part of the intervention: openness to experience (OE), behavioural awareness (BA), and valued action (VA).

## Data analysis

Descriptive statistical analyses were conducted for the feasibility and potential primary and secondary outcomes at baseline and each follow-up time point. Although this feasibility study was not powered to detect differences between groups in questionnaire-based outcomes, multi-level linear regression models were fitted to the data from each questionnaire to explore the direction and magnitude of effects. This method of analysis uses data from all time-points, and provides unbiased estimates, providing data can be assumed to be missing-at-random.

A cost-consequence analysis (CCA) was performed from a National Health Service (NHS) perspective. This analysis is commonly used in the literature to investigate the intervention's impact on costs and health outcomes in a feasibility study [42, 43]. The analysis listing the cost and outcomes associated with ACT (the treatment group) and the Usual Care Support Group (the control group) separately, allowing decision- makers to compare the relative value of the treatment options, and by calculating resources use and outcomes for each treatment. Regression analysis was used to adjust for patients' age, gender, baseline health-related quality of life, and body mass index. The intervention costs were calculated based on the activities involved in delivering the programmes such as staff salaries, administrative and managerial costs. Total

costs of the individuals in the two groups were estimated by multiplying the quantity of the resource use with the unit costs of health and social care in the NHS [44]. We presented the

**Table 2. Participant characteristics.**

| Columns by: Randomised group | SGC | ACT | Total |
|---|---|---|---|
| n (%) | 38 (47.5) | 42 (52.5) | 80 (100.0) |
| **Age,** mean (sd) | 51.3 (12.8) | 50.6 (11.6) | 50.9 (12.1) |
| **Sex,** n (%) | | | |
| *Female* | 31 (81.6) | 32 (76.2) | 63 (78.8) |
| *Male* | 7 (18.4) | 10 (23.8) | 17 (21.3) |
| **Marital Status,** n (%) | | | |
| *Civil Partnership* | 1 (2.7) | 0 (0.0) | 1 (1.3) |
| *Divorced* | 4 (10.8) | 3 (7.1) | 7 (8.9) |
| *Married* | 13 (35.1) | 19 (45.2) | 32 (40.5) |
| *Partner* | 7 (18.9) | 8 (19.0) | 15 (19.0) |
| *Partner–cohabiting* | 2 (5.4) | 0 (0.0) | 2 (2.5) |
| *Separated* | 0 (0.0) | 1 (2.4) | 1 (1.3) |
| *Single* | 9 (24.3) | 10 (23.8) | 19 (24.1) |
| *Widowed* | 1 (2.7) | 1 (2.4) | 2 (2.5) |
| *Missing* | 1 | | |
| **Employment status,** n (%) | | | |
| *F/T* | 22 (57.9) | 14 (33.3) | 36 (45.0) |
| *P/T* | 4 (10.5) | 5 (11.9) | 9 (11.3) |
| *Unemployed* | 5 (13.2) | 8 (19.0) | 13 (16.3) |
| *Homemaker* | 4 (10.5) | 5 (11.9) | 9 (11.3) |
| *Retired* | 2 (5.3) | 5 (11.9) | 7 (8.8) |
| *Student* | 0 (0.0) | 3 (7.1) | 3 (3.8) |
| *Other* | 1 (2.6) | 2 (4.8) | 3 (3.8) |
| **Surgery type,** n (%) | | | |
| *Band* | 1 (2.6) | 0 (0.0) | 1 (1.3) |
| *Bypass* | 9 (23.7) | 11 (26.2) | 20 (25.0) |
| *Bypass (reoperation from Sleeve)* | 0 (0.0) | 1 (2.4) | 1 (1.3) |
| *Gastric Band* | 0 (0.0) | 1 (2.4) | 1 (1.3) |
| *Gastric Sleeve* | 1 (2.6) | 0 (0.0) | 1 (1.3) |
| *Long limb or regular bypass* | 1 (2.6) | 1 (2.4) | 2 (2.5) |
| *Mini bypass* | 4 (10.5) | 4 (9.5) | 8 (10.0) |
| *Sleeve* | 19 (50.0) | 23 (54.8) | 42 (52.5) |
| *Sleeve & Bypass* | 0 (0.0) | 1 (2.4) | 1 (1.3) |
| *Sleeve (c&wm)* | 1 (2.6) | 0 (0.0) | 1 (1.3) |
| *Sleeve (revision from band)* | 1 (2.6) | 0 (0.0) | 1 (1.3) |
| *roux-en-y* | 1 (2.6) | 0 (0.0) | 1 (1.3) |
| **Height (cms),** mean (sd) | 165.6 (8.9) | 164.9 (9.3) | 165.2 (9.1) |
| **Highest weight (kgs),** mean (sd) | 134.9 (30.5) | 130.4 (31.1) | 132.6 (30.7) |
| **Pre-op weight (kgs),** mean (sd) | 127.3 (28.8) | 125.9 (30.9) | 126.5 (29.7) |
| **Weight at baseline (kgs),** mean (sd) | 95.7 (23.8) | 93.8 (27.0) | 94.7 (25.4) |
| **BMI at baseline,** mean (sd) | 34.8 (8.0) | 34.3 (8.6) | 34.5 (8.3) |

F/T, full time; P/T, part time; BMI, Body Mass index.

incremental costs (mean and standard deviation) associated with the intervention alongside the benefits (if any) identified in the trial when compared with the usual care group.

Interview data were analysed using a directed content analysis technique [45] with deductive coding. Transcripts were uploaded to NVivo 12 QSR [46]. Following familiarisation with the transcripts, the "Theoretical Framework of Acceptability of Healthcare Interventions" (TFA) [47] was used as the coding scheme. The TFA includes seven constructs of acceptability: affective attitude (i.e. how individuals feel about participating), perceived effectiveness (i.e. the extent to which the intervention is perceived to be likely to achieve its purpose), intervention coherence (i.e. participants' understanding of the intervention and how it works), ethicality (i.e. the fit of trial and intervention with an individual's values), burden (i.e. the perceived amount of effort required to participate), self-efficacy (i.e. participants' confidence that they can perform required behaviours) and opportunity costs (i.e. the extent to which benefits, profits, or values must be given up to participate). Data relating to the six ACT processes were also coded for ACT group participants and categorised under the 'perceived effectiveness' and 'intervention coherence' acceptability constructs.

Data relating to participation in the trial and SCG and in the ACT intervention were coded separately. Codes were agreed between the coding team (MB, HG, SB, EB) using one transcript before MB and EB coded all transcripts independently. Similarities and differences in views between the groups were highlighted and examples of disconfirming data were sought for each acceptability construct. The coding frequencies were presented as counts along with supporting quotes for each TFA component. Coding was discussed and agreed at multidisciplinary team meetings including the service user advisors.

This was a change from protocol where inductive thematic analysis was planned. However, given findings of successful recruitment but high attrition, it was felt that using a focused, structured approach to understand the acceptability, of both the trial processes and experience of the ACT group, would enable identification of specific areas to be addressed in future work. The TFA has been applied in this way to various interventions, including a men's mental health promotion program [48] a digital return to work intervention for common mental disorders [49] and an HIV differentiated care intervention for formerly incarcerated people re-entering community settings [50]. Its use therefore enhances the applicability of study findings by structuring findings according to recognised constructs of acceptability.

## Results

*Recruitment*: 264 participants were potentially eligible and screened. Of these 18 were ineligible (not fluent in English language), 166 did not participate (70 did not respond to contact, 59 unable to commit to sessions, 37 agreed but did not turn up for baseline appointment). Eighty participants were consented and randomised (Fig 1), consecutive participants were recruited following the 12 month follow up until equal numbers from each arm had been interviewed and data saturation was agreed to have occurred.

Participant characteristics are displayed in Table 2. There appeared to be good balance between the randomised groups on all characteristics.

### Data collection

This information is displayed in Fig 1. At 12 months, outcome data were available from 50% of those originally randomised to the SGC, and from 31% of those randomised to ACT.

Questionnaires took an average of 56 mins to complete at baseline when they were completed face to face and an average of 29 mins to complete at 12 months by phone (as at 3 and 6 months).

## Acceptability of the intervention

There were high levels of drop out from both groups (Table 3). Attendance was poor even at the first session, for both groups. Only 9 (29%) ACT participants completed $\geq$ half of the sessions, this was the case for 13 (35%) SGC participants. This means that it is unlikely that any participant received a 'dose' of ACT that would be expected to be sufficient to facilitate change. Our qualitative study explored the reasons for attrition, and findings are reported below.

## Process evaluation

Only six ACT participants competed the in-session measures of OE, BA, VA more than five times. Weekly changes were analysed over time. Results indicated that scores fluctuated throughout the intervention but there was no clear indication of improvement in any measure across the time points. No linear or non-linear patterns in the results could be identified.

## Exploration of primary and secondary outcome measures for a full RCT

Data from all questionnaires at each time point are included in the S1 Questionnaire data and include the levels of missing data. Changes from baseline and differences between groups: Findings from the multi-level linear regression models for the outcome measures are provided in S1 Table. Given that this was a feasibility study, not powered for efficacy, and the high levels of attrition, findings should be interpreted with caution. There were no statistically significant differences between groups and similar patterns across outcomes, but there are some interesting points to note. The mean change in weight from baseline to 12 months (Kgs) where a positive value represents weight gain and a negative value represents weight loss, was 2.6 (95% CI (-4.3 to 9.5); n = 13) and -1.4 (95% CI (-11.9 to 9.1; n = 10) in the SCG and ACT groups respectively (t-test for difference in means p = 0.47). However, considering individual responses, the number of participants who lost or maintained weight to within <2% of baseline was 8/13 (62%; 95% CI (32% to 86%) in SCG and 5/10 (50%; 95% CI (19% to 81%) in ACT, suggesting that average differences between groups were affected by large intra-individual variation in response (see S1 Table).

Health service use and costs: The ACT group had fewer hospital admissions (0 percent vs 13 percent), A&E visits (7.6 percent vs 20 percent), social worker visits (0 percent vs 7 percent), district nurse or health visitor visits (0 percent vs 6 percent), and visits to 'therapy in the

**Table 3. Acceptability of the intervention: number of participants who attended each session.**

|  | Group attendance | | Overall attendance |
| --- | --- | --- | --- |
| Session number | ACT | SGC | Total |
| 1 | 15 | 18 | 33 |
| 2 | 15 | 15 | 30 |
| 3 | 12 | 8 | 20 |
| 4 | 10 | 12 | 22 |
| 5 | 7 | 8 | 15 |
| 6 | 6 | 12 | 18 |
| 7 | 7 | 4 | 11 |
| 8 | 7 | 6 | 13 |
| 9 | 4 | 6 | 10 |
| 10 | 7 | 6 | 13 |

ACT, acceptance and commitment therapy; SCG, support group control.

community' (0 percent vs 6 percent) (0 percent vs 7 percent). They were more likely to visit an outpatient hospital (46 percent vs 26 percent), a GP (69 percent vs 40 percent) or a practise nurse (46 percent vs 27 percent), any telephone consultation (31 percent vs 20 percent), any day centre visit (38 percent vs 7 percent), a chiropractor (7 percent vs 0 percent), or any counselling, psychiatry, or community mental health visit (8 percent vs 7 percent). Except for any day centre visit ($p = 0.04$), no differences between groups were statistically significant (please see S1 File).

The cost of providing ACT services includes one hour of clinical psychologist time (£70) divided among ten patients. The training costs approximately £2.25 per patient (£450 per year for 200 patients). As a result, the total cost of providing ACT services is £72.25 per patient. SGC service delivery costs include one hour of assistant psychologist time (£34) divided among ten patients. This equates to £3.4 per patient or £34 for ten sessions. Last year, the ACT group spent £560 on medical expenses, while the usual care group spent £441. The intervention cost £633 for the ACT group and £475 for the usual/standard care group.

QALYs (Quality-adjusted Life Years): The unadjusted mean QALYs from baseline to 12 month follow-up were: ACT group = 0.61 (SD = 0.26); SGC = 0.59 SD = 0.37. The study only included 32 patients with baseline and 12-month EQ-5D measurements (19 in the usual care group and 13 in the ACT group). We then used a regression analysis to estimate the differences in QALY between two groups, adjusting for patients' age, gender, baseline health-related quality of life, and body mass index. The ACT group was associated with a 0.07 point reduction in QALY (P>0.05) in the adjusted analysis. The regression results were summarised in Table 4.

Table 5 summarised the results of the cost-consequences analysis. The cost of ACT intervention is £72.25 versus £34 in the usual care group. The estimated health-care cost in the 12 months following the intervention was £560, compared to £441 in the usual-care group over the previous 12 years. When compared to the control group, the ACT group was associated with a non-statistically significant reduction in QALY (-0.07, P>0.05).

## Acceptability: Qualitative findings

Eighteen participants were interviewed: nine from each arm (female n = 11). Three interviews were face to face and the subsequent 15 were by telephone; interview duration ranged from 35 to 65 minutes. Three of those interviewed did not attend any session (ACT n = 2, SGC n = 1), this was due to health reasons and being unable to attend due to other commitments. The 15 remaining participants interviewed attended from 1 to 9 sessions (ACT median = 5, SGC median = 8), though the ACT participants were able to 'catch up' on course content with the psychologist prior to the subsequent session.

**Coding results.**   All acceptability constructs were coded, except 'self efficacy' was not coded for the trial or SGC attendance, and though 'opportunity costs' was coded in relation to attending sessions it was not coded specifically in relation to participating in either group. Most codes related to 'perceived effectiveness' (50 in 7 interviews SGC, 46 in 6 interviews

**Table 4. Summary of regression output.**

|  | B | 95% CI | P-value |
| --- | --- | --- | --- |
| Baseline utility | 0.98 | (0.86,1.09) | 0.0005 |
| ACT group (ref usual care) | -0.07 | (-0.14, 0.006) | 0.072 |
| Age | 0.001 | (-0.001, 0.004) | 0.377 |
| Female (ref male) | -0.02 | (-0.116, 0.07) | -0.659 |
| Baseline BMI | -0.001 | (-0.005, 0.003) | 0.484 |

**Table 5. Costs and outcomes for the ACT and usual care groups.**

|  | ACT group | Usual care |
|---|---|---|
| Cost of Intervention | £72.25 | £34 |
| Cost of health service use | £560 | £441 |
| Total cost (Intervention + Service use) | £633 | £475 |
| Incremental QALY difference (ACT vs usual care) | -0.07 (95% CI = -0.14, 0.006) | |

ACT). Findings are presented as themes within each acceptability construct, supported by illustrative quotes (quotes are linked to a participant via participant number (1–80), gender (m/F) and the number of sessions attended out of 10).

**Affective attitude.**   All participants, except one, reported feeling positive at the time of enrolment about participating in the trial. The exception was someone who suggested that they felt obliged by their clinician to attend (9FA1); this person dropped out after the first session. The possibility that dissatisfaction with group allocation could explain the observed high attrition rates, particularly for the first session, was explored directly. Two participants declared a prior preference for the ACT group (19FA9, 20FC8). This worked out for one (19FA9), while the other was randomised to the SGC (20FC8). Nevertheless, this person participated in the trial and attended 8 out of 10 sessions. In addition, the groups were held on different days, and some participants (35FA0) stated being unavailable to attend their allocated group but would have attended on the other day. Group or timing preferences may therefore have been a factor contributing to low attendance for the first session.

*"I was a bit disappointed because I did want the other group. . ., mainly because it was on a more convenient night. . . . . . and the other reason was because it sounded more interesting"* [20FC8]

In terms of satisfaction with the groups, overall comments from participants of both arms were generally positive. However, participants from both the ACT group and the SGC reported disappointment with others' lack of attendance (Table 3). This was commonly cited as a reason for their own later non-attendance,

*"I said unless you could get at least a few more people turning up then, unfortunately, I would probably do the same [not turn up] 'cos it was a waste of my time."*[2MC3]

though others persevered despite sometimes being the only person there (e.g.39FC9). When participants received peer support, however, it was clearly valued, independent of allocation:

*"It was really nice to just um. . ..be in a roomful of people and be totally open and really honest and not reserved or holding back."*[20FC8]

"It was actually quite nice to sometimes meet other people, because on the sessions I meet up with other people. . .and we talked and shared stories and um er found out we were all in the same sort of boat really".[48MA6]

**Perceived effectiveness and intervention coherence.**   ACT group: More data were coded for the ACT group as participants were probed about ACT processes. All of the ACT processes were mentioned in some way by at least one participant. As might be expected, those who

attended more sessions appeared to have a greater understanding of the therapy, as more of their data could be coded to the ACT processes. No participant demonstrated full comprehension of every process, but increased awareness of thoughts and feelings and their impact on eating behaviour and the importance of choices being driven by values rather than internal experiences appeared to be understood:

> *"It made me aware that, by being aware of what you're doing and what was happening around you, it was very helpful [in showing you] what you enjoy then what you didn't enjoy."[59MA8]*

> *"I'm reaching for a biscuit tin, but I'm thinking to myself do I really need it? Then you think, actually I'm not hungry".[19FA9]*

All participants, except one who left after the first session (9FA1), reported benefits, including greater self-acceptance and behaviour changes:

> *"I came away with quite a few bits and pieces and er felt a lot better about myself".[56MA9]*

> *"I am not a great exerciser but doing small bits. . ..I have up it to 10 to 12,000 steps a day."[52FA9]*

Support Group Control: SGC participants understood that the groups were about sharing experiences. Several reported benefits of peer support, including gaining new insights:

> *"what came out was the fact that it was more of my head or even habit of eating. . . I think it was. . .listening to the others and what they used to eat and things like that. I mean it definitely helped.. . .".58FC8)*

There were no reports of changes in behaviour however.

**Ethicality.**   Participants' motivations for participating appeared in line with the intervention values. For instance, some looked forward to receiving psychological support, which they felt was both important and lacking in their treatment so far:

> *"I was keen to volunteer, because I think that's what is missing from the provisions of the support, the psychological aspects. Um because frankly it's the psychology that made me keep gaining weight".[47MC8]*

Others welcomed the chance for more information about how to maintain or lose weight (48MA6) especially in a group of people with shared experiences (7MC9):

> *"I haven't lost a great deal of weight. . . I thought it might have been a session where there's other people with the same the same problems and perhaps some experiences that I had had". [56MA9]*

A common motivation, which was outside the trial's explicit values, was a desire to help others, or to 'give back', in return for having been given surgery for which they felt grateful:

> *"to be able to pass on the things that you found were helpful."[&MC9]*

> *"I was so grateful for having being offered the surgery . . .I felt, well not obliged to, but I wanted to give back something."[39FC9]*

The extent to which they were able to do this, however, was reduced due to the limited group attendance meaning that opportunities to contribute were few.

**Burden.**   Principally, participants discussed difficulties travelling to the sessions, for instance time consuming journeys, transport problems or bad weather. Travel difficulties were linked to not being able to attend sessions which was a source of frustration for some:

*"I was quite annoyed that I didn't make um the last session but um it was to do with the trains".[58FC8]*

No additional burden specific to the SGC was coded. Some ACT participants found the homework burdensome, but, nevertheless, in line with the ACT model, committed to doing it:

*"[homework tasks were] tedious but I did all of them. . . I thought, if I'm committing to the sessions, then I'm going to do the homework, so I did, I did do that".[59FA8]*

**Self efficacy.**   This was not coded for participating in the trial or the SGC. ACT group participants referred to acquiring confidence to make their own choices, including one who felt empowered to stand up to their partner when making food choices:

*"to me like he [husband] was trying to sabotage it, because he was a was a feeder. . .and I think it made me realise that . . . I needed to work on relationships and making him understand. . ..".[19FA9]*

**Opportunity costs.**   Participants reported having to change their weekly schedule so that they could attend sessions and the issue of travel costs was raised as a potential barrier to attendance. No opportunity costs were coded specific to attending either the ACT group or SGC.

## Discussion

This trial tested the feasibility of delivering an RCT of ACT *versus* SCG. Overall, we found that the methods of the trial were feasible. All eligible participants were invited to participate, and we recruited and consented more than planned. Those who remained in the trial completed the outcome measures satisfactorily which allowed planned analyses to be conducted. However, the trial suffered from considerable attrition. This was linked to very low attendance at the groups which was similar for the ACT and SCG and was low even on the first session where only 57% attended.

It appeared that although data collection was hampered by lack of retention, other trial processes (i.e. recruitment, consent, data collection in those retained in the trial) would be feasible for a future trial but that the intervention, as delivered, may not be. We therefore decided to explore acceptability of the trial processes and intervention in detail using an established framework–The TFA [47]. This proved to be useful for capturing aspects of the trial and intervention which could be targeted to improve future trials. For instance, the acceptability constructs of 'burden' and 'opportunity costs' featured in interviews; participants cited session timing and a range of travel difficulties as reasons for non-attendance at initial and subsequent sessions. This is supported by studies investigating bariatric patients' attendance at routine follow up [51, 52], which has also been found to be low, with 'geographic distance' cited as a reason, along with family, professional and health problems. This suggests that future trials should make use of telehealth care tools to decrease the costs and burden of attendance.

Cognitive and behavioural telehealth interventions have been shown to increase attendance compared to in person delivery [51] and telehealth based behavioural interventions for obesity have been found not to be inferior to in-person interventions when assessed against weight gain [54].

Our interview data also suggested that the acceptability construct of 'ethicality' was important. This refers to the fit of trial and intervention with an individual's values. The intervention fitted with what the participants said they wanted (i.e. information and psychological support), and satisfaction with this was expressed as evidenced by the positive comments coded under the 'Affective Attitude' acceptability construct. However, joining the trial was commonly motivated by a desire to 'help others' or 'give back'. This, and another expressed desire, namely to meet other patients and share their experiences, was thwarted by the high levels of attrition in both groups This may explain why poor attendance at the first session led to reduced motivation to return, with subsequent drop out having a negative impact on future attendance.

Further, our finding that participants consented to join the study out of a motivation to help others, may suggest that they were not 'ready' for therapy themselves, an idea supported by our PPI advisors. The impact on weight loss of 'readiness' or motivation for bariatric surgery has been tested retrospectively [53] and prospectively [54], but no association with readiness to change scores or motivation such as changing appearance, fitness or medical concerns has been found. In other populations, the process of selecting patients suitable for therapy has also been studied with the aim of reducing attrition and improving outcomes, however the construct of readiness for therapy is complex and no established measures exist [55, 56]. Nevertheless "composite psychological variables" including combinations of increased self-esteem, coping skills and ability to use support have been found to predict weight loss in post-bariatric surgery through their effects on eating behaviour [57]. It is possible that patients approaching therapy from a purely altruistic point of view are not accepting of their own need for help and so their ability to use the support offered by therapy is reduced. Further trials of therapy for bariatric patients should include exploration of who responds best and why in order to inform future recruitment of suitable candidates.

With regard to perceived effectiveness and intervention coherence, few participants received a sufficient course of ACT for it to be meaningful. It is, therefore, unsurprising that no trends indicating a difference between the groups were identified. However, participants who attended the ACT groups reported some benefits and behaviour changes linked to ACT processes, especially regarding contacting the present moment, becoming more aware of behaviours and improved self efficacy. ACT group participants varied in their understanding of ACT processes, this appeared linked to attendance, but participants' description of the intervention was generally simplistic, with few able to give concrete examples of the processes covered in sessions and homework.

Nevertheless, this work has demonstrated that it is possible to deliver group ACT as part of routine care post bariatric care with minimal additional cost to services. Future trials could impact on large numbers of individuals and, if successful, reduce NHS costs significantly. For instance, in 2017/18 (latest available data NHS digital) there were 6,627 hospital admissions with a primary diagnosis of obesity and a main or secondary procedure of bariatric surgery; an increase of 2% on 2016/17 (6,492). However, approximately 20% of bariatric surgery patients regain lost weight with cost implications of £10,000 per re-operation as well as return of co-morbidities such as type-2 diabetes. If ACT is found, in future trials, to be effective at preventing weight regain, significant saving could be made. This study also highlights that the decision regarding the best primary and secondary outcome measures for a full RCT of ACT would require the careful consideration of many aspects of the proposed trial, including consensus in the most relevant patient-centred outcome to demonstrate efficacy. The data collected during

this feasibility study for a range of outcomes would be sufficient to inform the necessary sample size calculations for a future trial through estimates of standard deviation of the chosen outcome, but the final decision on the choice of primary endpoint should be taken by the research team following extensive consultation with patient representatives.

A strength of this feasibility trial is that, to reduce selection bias, we invited all eligible patients to participate. We noted early on that there was low attendance, so we ran additional iterations of the groups which confirmed that retention to the group rather than recruitment to the trial was an issue. We then explored acceptability of the trial and group processes with both high and low attenders using an established framework. This allowed us to pinpoint areas to address in a future trial to increase acceptability and retention. Nevertheless, due to the lack of attendance, few people received a meaningful level of the ACT intervention, so we are unable to recommend a definitive primary outcome for a future trial or suggest the processes by which ACT may work.

In conclusion, this study has demonstrated that an RCT of group ACT is feasible to deliver in bariatric patients post-surgery, but that the intervention is not feasible or acceptable as delivered. Low attendance at groups, due mainly to travel and scheduling difficulties, reduced acceptability as participants were unable to gain the peer support or to provide the help to others that had motivated them to join the trial. This suggests that acceptability, and hence retention, could be improved in a future full trial of effectiveness by using online intervention delivery whilst taking care to maximise participant satisfaction by providing opportunity for interaction with peers. The data from this trial also suggest the need for future research to explore the factors which influence when and which bariatric patients are most likely to benefit from psychological therapy and why in order to inform recruitment of suitable candidates to trials.

## Supporting information

**S1 Fig. CONSORT checklist.**
(PDF)

**S2 Fig. Individual weight changes.**
(PDF)

**S3 Fig. Results from mixed models–unadjusted.**
(PDF)

**S1 Dataset. FAB_fulldata_26SEP2022.**
(CSV)

**S1 Table. Questionnaire data.**
(PDF)

**S2 Table. Questionnaire results.**
(PDF)

**S3 Table. Results from multi-level linear regression models.**
(PDF)

**S1 File. Service use and costs.**
(PDF)

**S2 File. Feasibility of acceptance and commitment therapy for post-bariatric surgery patients: the FAB study protocol.**
(PDF)

## Acknowledgments

The authors would like to thank the study's PPI representatives, Georgina Hayman and Roy Ebbs, for bringing a service user perspective to every aspect of the study, including during regular Project Advisory Group meetings. We thank Ms Hayman for chairing this group.

## Author Contributions

**Conceptualization:** Elizabeth A. Barley, Samantha Scholtz.

**Data curation:** Marie Bovell, Kate Bennett-Eastley.

**Formal analysis:** Kate Bennett-Eastley, John Tayu Lee, Dayna Lee-Baggley, Simon S. Skene.

**Funding acquisition:** Elizabeth A. Barley, John Tayu Lee, Samantha Scholtz.

**Investigation:** Marie Bovell.

**Methodology:** Elizabeth A. Barley, John Tayu Lee, Samantha Scholtz.

**Project administration:** Marie Bovell.

**Supervision:** Elizabeth A. Barley, Samantha Scholtz.

**Visualization:** Kate Bennett-Eastley, Michael Z. Tai.

**Writing – original draft:** Elizabeth A. Barley.

**Writing – review & editing:** Elizabeth A. Barley, Marie Bovell, Kate Bennett-Eastley, John Tayu Lee, Dayna Lee-Baggley, Simon S. Skene, Michael Z. Tai, Sue Brooks, Samantha Scholtz.

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
