## [Decision Letter · Decision Letter 0]

17 Aug 2022

PONE-D-22-15868Addressing a critical need: A randomised controlled feasibility trial of Acceptance and Commitment Therapy for bariatric surgery patients at 15-18 months post-surgeryPLOS ONE

Dear Dr. Barley,

Thank you for submitting your manuscript to PLOS ONE. After careful consideration, we feel that it has merit but does not fully meet PLOS ONE’s publication criteria as it currently stands. Therefore, we invite you to submit a revised version of the manuscript that addresses the points raised during the review process.

We look forward to receiving your revised manuscript.

Kind regards,

Shahrad Taheri

Academic Editor

PLOS ONE

Journal Requirements:

2. We note that the original protocol that you have uploaded as a Supporting Information file contains an institutional logo. As this logo is likely copyrighted, we ask that you please remove it from this file and upload an updated version upon resubmission.

Reviewers' comments:

Reviewer's Responses to Questions

**Comments to the Author**

1. Is the manuscript technically sound, and do the data support the conclusions?

Reviewer #1: Yes

Reviewer #2: Partly

2. Has the statistical analysis been performed appropriately and rigorously? 

Reviewer #1: Yes

Reviewer #2: No

3. Have the authors made all data underlying the findings in their manuscript fully available?

Reviewer #1: No

Reviewer #2: No

4. Is the manuscript presented in an intelligible fashion and written in standard English?

Reviewer #1: Yes

Reviewer #2: Yes

5. Review Comments to the Author

Reviewer #1: The authors reported about the results of a randomized controlled feasibility trial to test the feasibility and acceptability of ACT following bariatric surgery a randomized controlled trial of 10 sessions of group ACT or Usual Care Support Group control (SGC) was delivered 15- 18 months post bariatric surgery they concluded, that the trial

processes were feasible, but that the ACT intervention was not acceptable as delivered. Our data suggest changes to recruitment and intervention delivery that would address this.

First of all I would like to congratulate the authors for providing and consistently reporting this feasibility study. I do have some minor comments to the presentation, which could be easily addressed in a review.

The appropriate guideline is the "CONSORT statement on Pilot and Feasibility Trials (see BMJ 2016; 355:i5239 http://dx.doi.org/10.1136/bmj. i5239)". Please use this checklist, although the difference appears to be minor. The extra statement on page 6 might not be necessary.

p5: Is it possible to give a variable, by which "Acceptability of the ACT group therapy" is assessed.

p7 Randomization. Please give more details, e.g. the randomization procedure, software (R software package) and how the random allocation was implemented. Please give more details with respect to blinding. Single blinding is not very specific. How is blinding implemented.

P7 I fully understand, that a strong sample size argument could not be given here. But when the primary endpoint is specified, a precision argument for estimation could be given - even post hoc - to illustrate the argument.

p8. Please include a table with the descriptive measures of the secondary outcome scores.

P10 or elsewhere: Could you derive some recommendation about the trails primary endpoint variable. Could you derive some ideas about the minimal detectable difference, i.e. the minimal difference to be detected resulting in a change of patient care?

p11: Please skip the p-values You may argue from the estimates of the difference and CI'S from the multilevel models

Rosenberger WF, Lachin JM. (2015) Randomization in Clinical Trials, Second Edition. Wiley, New York. DOI:10.1002/9781118742112

1. In general, p-values should be given with at least 3 decimal digits to avoid misinterpretation.

Reviewer #2: As the manuscript title states, this study addresses a critical need. Namely, the provision of psychological care to support weight management post-bariatric surgery. This is a well designed feasibility study that evaluates ACT group therapy delivered by a clinical psychologist in comparison to an unstructured support group delivered by a psychology graduate with no training in XX, in people who are 15-18 months post-bariatric surgery. There are some significant challenges in feasibility discovered in terms of intervention uptake and retention and the authors are transparent about this. However, there are some issues with the analysis and reporting of the study that should be improved prior to publication.

Introduction

The introduction needs revision – specifically in terms of the review of previous research on ACT for weight management. For example, the authors introduce ACT by stating that “Third wave cognitive behavioural therapies ….. are evidenced in the general population.14 , 15. What do the authors mean by “evidenced in the general population”? The two papers they cite are a (now out of date) review of mindfulness-based interventions by O'Reilly et al and a small trial of people with overweight and obesity from 2009 by Forman et al. The authors should more fully describe the literature to date in people with overweight and obesity. More recent and comprehensive systematic reviews have been conducted (e.g. Lawlor et al 2020), as have larger trials with longer term follow up. (eg Formal et al 2019 doi: 10.1002/oby.22412.)

The authors then cite a very small Iranian study in people with obesity. Is the point that the authors are trying to make that this is the only study in people with obesity? I suggest that this is not correct and that the authors should describe all relevant literature in people with obesity.

A more comprehensive review of literature to date for post bariatric surgery should then be reported. How does this study differ from the previous literature and what gap does it seek to fill?

The discussion should end with the aims of the study, rather than a summary of the rest of the paper.

The authors should also be careful to avoid self-plagiarism of their protocol paper http://dx.doi.org/10.18203/2349-3259.ijct20194655

Minor point - Person first language would mean not using overweight as something people “are”. I appreciate that some people find “people with obesity” to be grammatically difficult. Given that the real focus here is on obesity, I would simply remove the stats on overweight.

Methods

The methods are largely well described, and deviations from the protocol are highlighted.

I did not understand the justification for the intervention costings. It states that : There were no extra hours or treatment costs because the ACT group was led by a Band 7 psychologist as part of their regular duties. However, the control group did not receive care from a Band 7 psychologist whereas the support group was led by A Band 4 psychology graduate with no formal therapy training. The cost difference here would be substantial. It appears later in the results that these different costs are taken into account, so perhaps this section just needs rephrasing to be much clearer.

Minor – It states that Loss to follow up and group attrition were recorded in a consort diagram (Figure 1). I suspect that they were recorded somewhere else. They are presenting in the diagram.

Results

Large number of patients were screened as eligible but chose not to take part (with a substantial number unable to commit to the treatment sessions.

While I appreciate the granularity of Table 2, this makes it very difficult to follow. It would be more meaningful to present the number of all ACT and all SGT participants attending each session.

The authors include the following paragraph in the results “The decision regarding the best primary and secondary outcome measures for a full RCT would require the careful consideration of many aspects of the proposed trial, including consensus in the most relevant patient-centred outcome to demonstrate efficacy. The data collected during this feasibility study for a range of outcomes would be sufficient to inform the necessary sample size calculations for a future trial through estimates of standard deviation of the chosen outcome.” This should be moved to the discussion and the authors should be specific about HOW the current study is informative – what decisions does it help make? Can the authors design a trial based on this data? If so, what does this trial look like. If not, what further studies are needed to inform a trial?

The original intention in the protocol was to use inductive thematic analysis. The authors say that they used the Theoretical Framework of Acceptability (TFA) instead. They describe this as an “evidence-based” method. However, the TFA is not a method for qualitative data analysis. It is a framework for defining acceptability. How was the analysis conducted?

Tables 3 and 4 appear to be trying to quantify qualitative data and do not ad value to the paper. The way coding is described also suggests a very un-qualitative approach. It appears that they just “coded” each time something was said that was relevant to pre-determined codes and the results simply repeats “one person said this, two people said Y” with little to no attempt to interpret or synthesise the data. Phrases like “X was coded most frequently” suggest that this was not a qualitative analysis, but rather a labelling exercise. The qualitative analysis should be revisited and an appropriate method used and described.

Discussion

I suspect that the discussion will change following reanalysis of the qualitative data using appropriate qualitative methods. The discussion should be edited to be clearer where there are findings from this study and where the authors are referring to findings from the general literature. Ideally, we should be able to discern the novel contribution of this study.

The strengths and limitations of the study should be clearly discussed

6. PLOS authors have the option to publish the peer review history of their article (what does this mean?). If published, this will include your full peer review and any attached files.

Reviewer #1: No

Reviewer #2: No

---

## [Author Response · Author response to Decision Letter 0]

4 Oct 2022

PLOS One – FAB paper: Response to Reviewers

Journal Requirements: When submitting your revision, we need you to address these additional requirements:

Author response: we apologise for this error and trust that this is now satisfactory.

• 2. We note that the original protocol that you have uploaded as a Supporting Information file contains an institutional logo. As this logo is likely copyrighted, we ask that you please remove it from this file and upload an updated version upon resubmission. 

Author response: logo removed as requested

• 3. In your Data Availability statement, you have not specified where the minimal data set underlying the results described in your manuscript can be found. PLOS defines a study's minimal data set as the underlying data used to reach the conclusions drawn in the manuscript and any additional data required to replicate the reported study findings in their entirety. All PLOS journals require that the minimal data set be made fully available. For more information about our data policy, please see http://journals.plos.org/plosone/s/data-availability. Upon re-submitting your revised manuscript, please upload your study’s minimal underlying data set as either Supporting Information files or to a stable, public repository and include the relevant URLs, DOIs, or accession numbers within your revised cover letter. For a list of acceptable repositories, please see http://journals.plos.org/plosone/s/data-availability#loc-recommended-repositories. Any potentially identifying patient information must be fully anonymized.

Author response: uploaded as requested

Author response: N/A

Author response: we note in our cover letter that the minimal dataset is uploaded as a supplementary file.

Author response: now added (p34)

Reviewer comments: 

Reviewer #1: The authors reported about the results of a randomized controlled feasibility trial to test the feasibility and acceptability of ACT following bariatric surgery a randomized controlled trial of 10 sessions of group ACT or Usual Care Support Group control (SGC) was delivered 15- 18 months post bariatric surgery they concluded, that the trial processes were feasible, but that the ACT intervention was not acceptable as delivered. Our data suggest changes to recruitment and intervention delivery that would address this.

First of all I would like to congratulate the authors for providing and consistently reporting this feasibility study. I do have some minor comments to the presentation, which could be easily addressed in a review.

Author Response: We thank this reviewer for their thoughtful comments. 

Reviewer #1: The appropriate guideline is the "CONSORT statement on Pilot and Feasibility Trials (see BMJ 2016; 355:i5239 http://dx.doi.org/10.1136/bmj. i5239)". Please use this checklist, although the difference appears to be minor. The extra statement on page 6 might not be necessary.

Author Response: Thank you this version has now been used (S1)

p5: Is it possible to give a variable, by which "Acceptability of the ACT group therapy" is assessed.

Author Response: we have now added “Acceptability of the intervention was assessed as group attendance and through interview data.”

Reviewer #1: p7 Randomization. Please give more details, e.g. the randomization procedure, software (R software package) and how the random allocation was implemented. Please give more details with respect to blinding. Single blinding is not very specific. How is blinding implemented.

Author Response: Thank you, we have now added details to P7 under ‘randomization’

Reviewer #1: P7 I fully understand, that a strong sample size argument could not be given here. But when the primary endpoint is specified, a precision argument for estimation could be given - even post hoc - to illustrate the argument.

Author Response: Thank you we have now noted the precision using confidence intervals of the main feasibility outcomes and added an illustrative post-hoc argument to justify the sample size.

Reviewer #1: p8. Please include a table with the descriptive measures of the secondary outcome scores.

Author Response: Thank you for this, all this information can be found in the Supplementary materials.

Reviewer #1: P10 or elsewhere: Could you derive some recommendation about the trails primary endpoint variable. Could you derive some ideas about the minimal detectable difference, i.e. the minimal difference to be detected resulting in a change of patient care?

Author Response: We have moved the comments about the primary endpoint from the results section to the discussion section (as requested by reviewer 2), and added some additional comments. The study was not designed to detect minimal detectable differences for validated outcomes, but sufficient data exists to support a sample size calculation.

Reviewer #1: p11: Please skip the p-values You may argue from the estimates of the difference and CI'S from the multilevel models. Rosenberger WF, Lachin JM. (2015) Randomization in Clinical Trials, Second Edition. Wiley, New York. DOI:10.1002/9781118742112

Author Response: Within the manuscript, the p-values have now been removed and replaced by estimated differences and confidence intervals as suggested.

Reviewer #1: In general, p-values should be given with at least 3 decimal digits to avoid misinterpretation.

Author Response: please see above response

Reviewer #2: As the manuscript title states, this study addresses a critical need. Namely, the provision of psychological care to support weight management post-bariatric surgery. This is a well designed feasibility study that evaluates ACT group therapy delivered by a clinical psychologist in comparison to an unstructured support group delivered by a psychology graduate with no training in XX, in people who are 15-18 months post-bariatric surgery. There are some significant challenges in feasibility discovered in terms of intervention uptake and retention and the authors are transparent about this. However, there are some issues with the analysis and reporting of the study that should be improved prior to publication.

Reviewer #2: Introduction

The introduction needs revision – specifically in terms of the review of previous research on ACT for weight management. For example, the authors introduce ACT by stating that “Third wave cognitive behavioural therapies ….. are evidenced in the general population.14 , 15. What do the authors mean by “evidenced in the general population”? The two papers they cite are a (now out of date) review of mindfulness-based interventions by O'Reilly et al and a small trial of people with overweight and obesity from 2009 by Forman et al. The authors should more fully describe the literature to date in people with overweight and obesity. More recent and comprehensive systematic reviews have been conducted (e.g. Lawlor et al 2020), as have larger trials with longer term follow up. (eg Formal et al 2019 doi: 10.1002/oby.22412.)

The authors then cite a very small Iranian study in people with obesity. Is the point that the authors are trying to make that this is the only study in people with obesity? I suggest that this is not correct and that the authors should describe all relevant literature in people with obesity.

A more comprehensive review of literature to date for post bariatric surgery should then be reported. How does this study differ from the previous literature and what gap does it seek to fill?

Author Response: We have now removed those references, updated the background section and made clearer what this study adds. 

Reviewer #2: The discussion should end with the aims of the study, rather than a summary of the rest of the paper.

Author Response: thank you, we have now changed this.

Reviewer #2: The authors should also be careful to avoid self-plagiarism of their protocol paper http://dx.doi.org/10.18203/2349-3259.ijct20194655

Author Response: thank you, we have reviewed the paper for self-plagarism

Reviewer #2: Minor point - Person first language would mean not using overweight as something people “are”. I appreciate that some people find “people with obesity” to be grammatically difficult. Given that the real focus here is on obesity, I would simply remove the stats on overweight.

Author Response: we have removed the stats on overweight and used the term people living with obesity throughout. 

Reviewer #2: Methods

The methods are largely well described, and deviations from the protocol are highlighted.

I did not understand the justification for the intervention costings. It states that : There were no extra hours or treatment costs because the ACT group was led by a Band 7 psychologist as part of their regular duties. However, the control group did not receive care from a Band 7 psychologist whereas the support group was led by A Band 4 psychology graduate with no formal therapy training. The cost difference here would be substantial. It appears later in the results that these different costs are taken into account, so perhaps this section just needs rephrasing to be much clearer.

Author Response: thank you, we have removed reference to no extra hours/treatment costs as we agree this was confusing.

Reviewer #2: Minor – It states that Loss to follow up and group attrition were recorded in a consort diagram (Figure 1). I suspect that they were recorded somewhere else. They are presenting in the diagram.

Author Response: thank you, we have corrected our language accordingly.

Reviewer #2: Results

Large number of patients were screened as eligible but chose not to take part (with a substantial number unable to commit to the treatment sessions.

While I appreciate the granularity of Table 2, this makes it very difficult to follow. It would be more meaningful to present the number of all ACT and all SGT participants attending each session.

Author Response: we have now changed Table 2 accordingly and agree that it is clearer.

Reviewer #2: The authors include the following paragraph in the results “The decision regarding the best primary and secondary outcome measures for a full RCT would require the careful consideration of many aspects of the proposed trial, including consensus in the most relevant patient-centred outcome to demonstrate efficacy. The data collected during this feasibility study for a range of outcomes would be sufficient to inform the necessary sample size calculations for a future trial through estimates of standard deviation of the chosen outcome.” This should be moved to the discussion and the authors should be specific about HOW the current study is informative – what decisions does it help make? Can the authors design a trial based on this data? If so, what does this trial look like. If not, what further studies are needed to inform a trial?

Author Response: We have moved the comments about the primary endpoint from the results section to the discussion section as requested and improved the discussion. The study was not designed to detect minimal detectable differences for validated outcomes, but sufficient data exists to support a sample size calculation.

Reviewer #2: The original intention in the protocol was to use inductive thematic analysis. The authors say that they used the Theoretical Framework of Acceptability (TFA) instead. They describe this as an “evidence-based” method. However, the TFA is not a method for qualitative data analysis. It is a framework for defining acceptability. How was the analysis conducted?

Tables 3 and 4 appear to be trying to quantify qualitative data and do not ad value to the paper. The way coding is described also suggests a very un-qualitative approach. It appears that they just “coded” each time something was said that was relevant to pre-determined codes and the results simply repeats “one person said this, two people said Y” with little to no attempt to interpret or synthesise the data. Phrases like “X was coded most frequently” suggest that this was not a qualitative analysis, but rather a labelling exercise. The qualitative analysis should be revisited and an appropriate method used and described.

Author Response: thank you for this. it appears that in trying to be concise we left out important information. We have now explained and referenced our methods in more detail, cut tables 3&4 and incorporated quotes into the text.

Reviewer #2: 

Discussion

I suspect that the discussion will change following reanalysis of the qualitative data using appropriate qualitative methods. The discussion should be edited to be clearer where there are findings from this study and where the authors are referring to findings from the general literature. Ideally, we should be able to discern the novel contribution of this study.

The strengths and limitations of the study should be clearly discussed

Author Response: We have now changed the discussion and addressed these issues and hope that it is now clearer.

---

## [Decision Letter · Decision Letter 1]

27 Jan 2023

PONE-D-22-15868R1Addressing a critical need: A randomised controlled feasibility trial of Acceptance and Commitment Therapy for bariatric surgery patients at 15-18 months post-surgeryPLOS ONE

Dear Dr. Barley,

Thank you for submitting your manuscript to PLOS ONE. After careful consideration, we feel that it has merit but does not fully meet PLOS ONE’s publication criteria as it currently stands. Therefore, we invite you to submit a revised version of the manuscript that addresses the points raised during the review process.

The manuscript has been evaluated by two reviewers, and their comments are available below.

The reviewers have raised a number of concerns that need attention. They request additional information on methodological aspects of the study (e.g they find that the cost consequence method is not well described), and question why less than 50% of data was completed (trail retention issues?). Could you please revise the manuscript to carefully address the concerns raised?

We look forward to receiving your revised manuscript.

Kind regards,

Katrien Janin

Staff Editor

PLOS ONE

Journal Requirements:

Reviewers' comments:

Reviewer's Responses to Questions

**Comments to the Author**

1. If the authors have adequately addressed your comments raised in a previous round of review and you feel that this manuscript is now acceptable for publication, you may indicate that here to bypass the “Comments to the Author” section, enter your conflict of interest statement in the “Confidential to Editor” section, and submit your "Accept" recommendation.

Reviewer #1: All comments have been addressed

Reviewer #2: All comments have been addressed

2. Is the manuscript technically sound, and do the data support the conclusions?

Reviewer #1: Yes

Reviewer #2: Partly

3. Has the statistical analysis been performed appropriately and rigorously? 

Reviewer #1: Yes

Reviewer #2: I Don't Know

4. Have the authors made all data underlying the findings in their manuscript fully available?

Reviewer #1: No

Reviewer #2: Yes

5. Is the manuscript presented in an intelligible fashion and written in standard English?

Reviewer #1: Yes

Reviewer #2: Yes

6. Review Comments to the Author

Reviewer #1: (No Response)

Reviewer #2: The authors have done a reasonable job of responding to my earlier comments.

The qualitative analysis is still a little thin, but I think that is the nature of the data collected and is unlikely to improve.

Some further minor edits are suggested prior to publication

The following sentence does not make sense: Patients who expressed suicidal ideation (score >0 on 125 Q9, PHQ-9) or at psychiatry assessment [26], were unable to communicate in English or to commit to 126 attendance were excluded.

the cost consequence method is not well described. What was included in the analysis? How was this done?.

It is not clear how the ACT group was associated with a 0.07 QALY reduction in mean incremental QALY if mean QALYs accrued from baseline to 12 month 346 follow-up were: ACT (n=13) = 0.61 SD 0.26; SGC (n=19) = 0.59 SD 0.37.

The authors comment that data collection was feasible and acceptable, but less than 50% completed 12 month data collection. Tis suggests problems with data collection and trial retention.

7. PLOS authors have the option to publish the peer review history of their article (what does this mean?). If published, this will include your full peer review and any attached files.

Reviewer #1: No

Reviewer #2: No

---

## [Author Response · Author response to Decision Letter 1]

14 Feb 2023

Katrien Janin

Staff Editor

PLOS ONE

Dear Dr Janin

Thank you for your email. We have now revised the manuscript in accordance with the new minor points that Reviewer 2 has made in their second review. The changes are listed below. 

We hope that the manuscript is now acceptable for publication. 

Reviewer #1: (No Response)

Author response: we thank Reviewer 1 for their positive review.

Reviewer #2: The authors have done a reasonable job of responding to my earlier comments.

The qualitative analysis is still a little thin, but I think that is the nature of the data collected and is unlikely to improve.

Author response: we thank Reviewer 2 for their very detailed reviews. We agree with the above comment. 

Reviewer 2: Some further minor edits are suggested prior to publication

The following sentence does not make sense: Patients who expressed suicidal ideation (score >0 on 125 Q9, PHQ-9) or at psychiatry assessment [26], were unable to communicate in English or to commit to 126 attendance were excluded.

Author Response: Thank you, we have restructured the sentence accordingly: “Patients who expressed suicidal ideation (score >0 on Q9 of the PHQ-9 [26] or at psychiatry assessment), or who were unable to communicate in English, or who were unable to commit to attendance were excluded.”

Reviewer 2: the cost consequence method is not well described. What was included in the analysis? How was this done?

Author Response: The requested detail has now been added with references (p11)

Reviewer 2: It is not clear how the ACT group was associated with a 0.07 QALY reduction in mean incremental QALY if mean QALYs accrued from baseline to 12 month 346 follow-up were: ACT (n=13) = 0.61 SD 0.26; SGC (n=19) = 0.59 SD 0.37.

Author Response: this as now been clarified (p18-19)

Reviewer 2: The authors comment that data collection was feasible and acceptable, but less than 50% completed 12 month data collection. Tis suggests problems with data collection and trial retention.

Author Response: it is unclear to where in the manuscript the reviewer is referring, we have been clear about trial retention issues throughout, we have now adapted the following existing sentence (p25) to emphasise the point: “It appeared that although data collection was hampered by lack of retention, other trial processes (i.e. recruitment, consent, data collection in those retained in the trial) would be feasible for a future trial but that the intervention, as delivered, may not be.”

Editor comment: Any changes to the reference list should be mentioned in the rebuttal letter that accompanies your revised manuscript. If you need to cite a retracted article, indicate the article’s retracted status in the References list and also include a citation and full reference for the retraction notice.

Author Response: 3 references have now been added to support the additional information requested about the cost-consequence analysis:

44. National Institute for Health and Care Excellence. Medical technologies evaluation programme methods guide 2017. Available at: https://www.nice.org.uk/process/pmg33/chapter/evidence-synthesis-and-cost-consequence-analysis. Accessed 9 Feb 2023. 

45. Drummond M. Methods for the economic evaluation of Health Care Programmes. Oxford, United Kingdom: Oxford University Press; 2015. 

46. Curtis LA, Burns A. Unit Costs of Health and Social Care 2018 [Internet]. University of Kent; 2018. Available from: https://kar.kent.ac.uk/id/eprint/70995

Thank you for your consideration, we look forward to your response. 

kind regards

Elizabeth Barley PhD CPsychol AFBPsS RGN PGCAP FHEA

Professor of Mental Health Sciences and Nursing

School of Health Sciences 

University of Surrey

30 Priestley Road

Surrey Research Park

Guildford

Surrey

GU2 7YH

---

## [Editor Report · Decision Letter 2]

24 Feb 2023

Addressing a critical need: A randomised controlled feasibility trial of Acceptance and Commitment Therapy for bariatric surgery patients at 15-18 months post-surgery

PONE-D-22-15868R2

Dear Dr. Barley,

Thank you for submitting your manuscript to PLOS ONE; I sincerely apologise for the unusually delayed review timeframe. We’re pleased to inform you that your manuscript has been judged scientifically suitable for publication and will be formally accepted for publication once it meets all outstanding technical requirements.

Kind regards,

Emily Chenette

Editor in Chief

PLOS ONE
---

## [Editor Report · Acceptance letter]

28 Feb 2023

PONE-D-22-15868R2 

Addressing a critical need: A randomised controlled feasibility trial of Acceptance and Commitment Therapy for bariatric surgery patients at 15-18 months post-surgery 

Dear Dr. Barley:

I'm pleased to inform you that your manuscript has been deemed suitable for publication in PLOS ONE. Congratulations! Your manuscript is now with our production department. 

Kind regards, 

on behalf of

Dr Emily Chenette 

Staff Editor

PLOS ONE